

# Biosynthesis of silver nanoparticles using *Caesalpinia ferrea* (Tul.) Martius extract: physicochemical characterization, antifungal activity and cytotoxicity

Mônica R. P. S. Soares[1,*], Rafael O. Corrêa[1,*], Pedro Henrique F. Stroppa[2,*], Flávia C. Marques[2,*], Gustavo F. S. Andrade[2,*], Charlane C. Corrêa[2,*], Marcos Antônio F. Brandão[1,*] and Nádia R. B. Raposo[1,*]

[1] Núcleo de Pesquisa e Inovação em Ciências da Saúde (NUPICS)—Faculdade de Farmácia, Universidade Federal de Juiz de Fora, Juiz de Fora, Minas Gerais, Brazil

[2] Departamento de Química, Universidade Federal de Juiz de Fora, Juiz de Fora, Minas Gerais, Brazil

* These authors contributed equally to this work.

Corresponding author
Nádia R. B. Raposo,
nadiacritt@gmail.com

## ABSTRACT

**Background:** Green synthesis is an ecological technique for the production of well characterized metallic nanoparticles using plants. This study investigated the synthesis of silver nanoparticles (AgNPs) using a *Caesalpinia ferrea* seed extract as a reducing agent.

**Methods:** The formation of AgNPs was identified by instrumental analysis, including ultraviolet–visible (UV–Vis) spectroscopy, scanning electron microscopy (SEM), X-ray diffraction (XRD) of the AgNPs, and surface-enhanced Raman scattering (SERS) spectra of rhodamine-6G (R6G). We studied the physicochemical characterization of AgNPs, evaluated them as an antifungal agent against *Candida albicans*, *Candida kruzei*, *Candida glabrata* and *Candida guilliermondii*, and estimated their minimum inhibitory concentration (MIC) and minimum fungicidal concentration (MFC) values. Lastly, this study evaluated the cytotoxicity of the AgNPs in murine L929 fibroblasts cells using an MTT assay.

**Results:** The UV–Vis spectroscopy, SERS, SEM and XRD results confirmed the rapid formation of spheroidal 30–50 nm AgNPs. The MIC and MFC values indicated the antifungal potential of AgNPs against most of the fungi studied and high cell viability in murine L929 fibroblasts. In addition, this study demonstrated that *C. ferrea* seed extracts may be used for the green synthesis of AgNPs at room temperature for the treatment of candidiasis.

# INTRODUCTION

Antimicrobial resistance is a phenomenon of infectious microbial flora that aids them in resisting antimicrobial agents to which they were previously sensitive (*Lakum, Shah & Chikhalia, 2014*). The occurrence of antifungal multi-drug resistant infections, which

affects the public health worldwide, has been increasing to alarming levels and has stimulated investigations on plant species for the treatment of candidiasis. Microorganisms of the genus *Candida* have acquired resistance to a variety of antifungal drugs, leading to poor treatment and increasing disease severity. Hence, studies on new pharmacologically active compounds for the treatment of fungal conditions are warranted (*Morais-Braga et al., 2016*).

Recent advances in nanoscience and nanotechnology have radically changed the way we diagnose, treat and prevent various diseases in all aspects of human life. Silver nanoparticles (AgNPs) are one of the most vital nanomaterials among several metallic nanoparticles that are involved in biomedical applications (*Wei et al., 2015*; *Zhang et al., 2016*). They also show antimicrobial activity making them applicable to different areas of medicine with the potential to combat the proliferation of microorganisms and yeasts (*Dúran et al., 2010*; *Iravani, 2011*; *Vasquez-Munoz, Avalos-Borja & Castro-Longoria, 2014*; *Jacometo et al., 2015*; *Szweda et al., 2015*; *Das et al., 2016*). However, there is a growing need for developing environmentally friendly processes of synthesis of nanoparticles that do not use toxic chemicals (*Song & Kim, 2009*).

Biological methods of synthesis of nanoparticles using microorganisms (*Castro-Longoria, Vilchis-Nestor & Avalos-Borja, 2011*; *Geronikaki et al., 2013*; *Xue et al., 2016*), enzymes (*Jacometo et al., 2015*), and plant or plant extracts (*Iravani, 2011*; *Mallmann et al., 2015*; *Kubyshkin et al., 2016*; *Shaik et al., 2017*) have been suggested as ecofriendly alternatives to chemical and physical methods. Using plant extracts for synthesis of nanoparticles can be advantageous over other biological processes by eliminating the elaborate process of maintaining cell cultures (*Iravani, 2011*). It can also be suitably scaled up for large-scale synthesis of nanoparticles (*Song & Kim, 2009*). Plant species have been used for biosynthesis of nanoparticles to preserve the antimicrobial activity of AgNPs, reducing its toxic effects on human cell lines, and increasing its practical application without impacting on the environment (*Dúran et al., 2010*; *Iravani, 2011*). Plant extracts function as bioreducers besides having nanoparticle stabilizers in colloidal solutions of metals, such as silver and gold. AgNPs are easy to prepare and chemically modified by reduction of photochemical, radiolytic, or biogenic ions or formations (*Xue et al., 2016*).

Brazil has an advantage in this market because it has the greatest biodiversity in the world and a genetic heritage of great potential for the development of new drugs (*Yunes, Pedrosa & Cechinel Filho, 2001*; *Dutra et al., 2016*). In the Amazon, there are approximately 25–30,000 species, but less than 1% of the Brazilian plant species have been was investigated from the chemical and pharmacological point of view (*Yunes, Pedrosa & Cechinel Filho, 2001*; *Araújo et al., 2014*; *Oliveira et al., 2014*; *Kobayashi et al., 2015*; *Dutra et al., 2016*).

Among the plant species of the Brazilian biome, *Caesalpinia ferrea* (Tul.) Martius is popularly referred to as ironwood or jucá, and the traditional pharmacological applications of shells, seeds, roots and fruits from this species are used to heal wounds and bruises (*Cavalheiro et al., 2009*; *Bariani et al., 2012*; *Silva et al., 2015*). Accordingly, anti-inflammatory (*Araújo et al., 2014*; *Kobayashi et al., 2015*), antifungal (*Bariani et al., 2012*), antihistaminic, antiallergic, anticoagulant and larvicidal activities

(*Cavalheiro et al., 2009*), and antiproliferative, cytoprotective and antimutagenic effects have been shown (*Silva et al., 2015*).

In this research, we investigated the synthesis of AgNPs using a *C. ferrea* seed extract as a reducing agent for the treatment of candidiasis. The formation of AgNPs was identified by instrumental analysis and results confirmed the rapid formation of spheroidal 30–50 nm AgNPs. In addition, the impact of green-fabricated AgNPs on *Candida* spp. and cell viability in murine L929 fibroblast cells was assessed.

# MATERIALS AND METHODS

## Preparation of the seed extract

Seeds of *C. ferrea* (Tul.) Martius were acquired at the "Ver-O-Peso" market in Belem, Pará, Brazil (Latitude 01270210 0S and Longitude 48300160 OW). Extract was obtained by macerating 25 g seeds in 70% (v/v) ethanol for 72 h in the dark. Extract was filtered under low pressure (Rotavapor® R-210; Buchi, Flawil, Switzerland), were lyophilized (ALPHA 1-4 ID plus; Christ, Osterode am Harz, Germany) under 1.8 mBar pressure at −14 °C, and was then maintained at room temperature in the dark until use.

## Biosynthesis of AgNPs with extract of *C. ferrea*

Stock solution of *C. ferrea* extract was prepared by solubilizing 50 mg of lyophilized powder in 50 mL of ethanol 70% (v/v), and then it was sonicated for 10 min. For the formation of AgNPs by green synthesis, 10 μL of *C. ferrea* stock solution was added into 990 μL of a silver nitrate solution (3 mM—$AgNO_3$—Synth, São Paulo, Brazil) and 0.5 μL of sodium hydroxide (5 M—NaOH—Synth, São Paulo, Brazil). The reaction mixture passed light yellow to color brown immediately indicating the formation of the AgNPs, the process was carried out at room temperature.

In order to confirm the formation of the AgNPs, measurements of absorption of reaction aliquots were performed immediately after and at intervals of 12–96 h using a spectrometer (Multiskan GO Microplate Spectrophotometer, Toronto, Ontario, Canada). All measurements showed maximum absorption near 423 nm (see Fig. 1).

## Nanoparticle characterization
### Ultraviolet–visible spectroscopy

Ultraviolet–visible (UV–Vis) spectra were obtained by a Shimadzu UV-1800 spectro-photometer with 1 cm path length quartz cell. After mixing the seed *C. ferrea* extract with $AgNO_3$ and NaOH solutions, the color changed from yellow to brown. The formation of AgNPs was determined by UV–Vis measurements, where aliquots of the reaction, see section "Biosynthesis of AgNPs with extract of *C. ferrea*," were analyzed at intervals of every 12 h for 96 h. UV–Vis spectroscopy showed a Plasmon resonance centered at 423 nm (Fig. 1).

### Scanning electron microscopy

Scanning electron microscopy (SEM) measurements were obtained using a FEI microscope, model Quanta 250, working at 25 or 30 kV in the W-filament. The colloidal

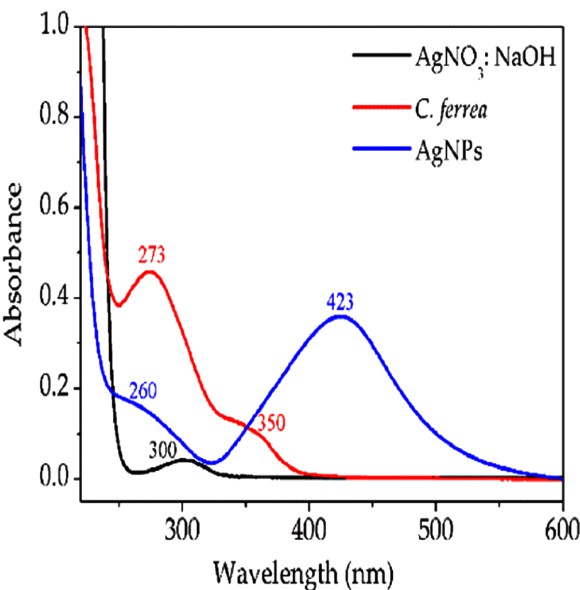

**Figure 1 UV–Vis absorption spectra.** Shows UV–Vis absorption spectra in the region between 200 and 600 nm for AgNO₃ solution, *C. ferrea* extract and AgNPs. A total of 24 h post-exposure.

solution was deposited on a silicon slide and subsequently dried in a vacuum oven before SEM measurements.

### X-ray diffraction of the AgNPs

X-ray diffraction (XRD) experiments were carried out in a Bruker diffract meter, D8 Advance DaVinci model with Cu Kα radiation (40 kV, 40 mA). The colloidal solutions were mixed with amorphous silica microbeads followed by centrifugation. The precipitate was dried out under vacuum. The resulting powder was spread out on a silicon antireflection slide.

### Surface-enhanced Raman scattering spectrum of R6G

Raman spectra were measured using a Bruker Raman spectrometer; model Senterra with laser excitation at 633 nm, and laser power at 10 mW. Spectral data were collected using a 50× microscope objective (NA = 0.51) with 30 s integration time. The surface-enhanced Raman scattering (SERS) samples were prepared by mixing 360 μL of colloidal solution with 40 μL of aqueous solutions of the probe molecule, resulting in a final R6G concentration of $1.0 \times 10^{-5}$ mol L$^{-1}$.

## Antifungal activity of AgNPs
### Fungal species

Microbiological analyses were performed using standard *C. albicans*, *American Type Culture Collection* (ATCC) 10231; *C. glabrata* (Taniwaki, M.H.), Collection of Tropical Cultures (CCT) 0728; *Candida krusei*, (FTI) CCT 1517; *C. guilliermondii* (CCT), 1890 from the Foundation André Tosello (Campinas, São Paulo, Brazil).

*Antifungal activity*

Minimum inhibitory concentration (MIC) and minimum fungicidal concentration (MFC) values were established according to the protocol M27-A2 of the *Clinical and Laboratory Standards Institute (CLSI) (2002)*. Briefly, fungal suspensions were obtained from respective lineages using sterile saline solution (0.9% (v/v)) after 48 h of growth at $35 \pm 2$ °C. Densities of suspensions were adjusted to 89–90% transmittance in a spectrophotometer (Libra S12; Biochrom, Copenhagen, Denmark) at a fixed wavelength of 530 nm. Samples were then diluted in buffered Roswell Park Memorial Institute (RPMI) 1640 medium (Sigma Aldrich Chemistry, Burlington, MA, USA) with 3 (N-morpholino) propanesulfonic (MOPS) (JTBaker, Phillipsburg, New Jersey, USA) to obtain $5$–$25 \times 10^2$ CFU and pH was adjusted to $7.0 \pm 0.1$ using 0.5M sodium hydroxide.

Assays were performed in triplicate and plant extract and AgNPs were diluted to 9.7–5,000 $\mu g$ $mL^{-1}$ in RPMI-1640 medium containing MOPS buffer and 2 $\mu L$ of 70% (v/v) ethanol solution.

Sterile polypropylene microplates with 96 wells and level depths were purchased from Sarstedt (Germany), and 100 $\mu L$ aliquots of respective dilutions of vegetable extract and AgNPs were added with 100 $\mu L$ aliquots of standardized fungal suspensions. Control wells contained 100 $\mu L$ of the same inoculated culture medium containing 2 $\mu L$ of 70% ethanol and 98 $\mu L$ of RPMI 1640 medium containing MOPS. The negative control contained 200 $\mu L$ of medium. Amphotericin B (Cristália, Itapira, São Paulo, Brazil) and nystatin (Cavalieri, Juiz de Fora, Minas Gerais, Brazil) were used as reference drugs at 0.0313–16.0 $\mu g$ $mL^{-1}$.

After inoculation, microplates were incubated at $35 \pm 2$ °C for 48 h, and MIC values were established as the lowest concentration in which no fungal growth was observed. MFC values were determined using the microdilution method. Briefly, 10 $\mu L$ aliquots were withdrawn from wells in which no growth was observed in the MIC procedure, were transferred to new wells containing 1 mL of Sabouraud dextrose broth (SDB), and were incubated at $35 \pm 2$ °C for 48 h. Subsequently, concentrations with no fungal growth were classified as MFC values.

## Cytotoxicity

*Evaluation of cell viability using MTT assays*

Cell viability of murine L929 fibroblasts cells (Federal University of Minas Gerais) was evaluated after culture in DMEM (Nutricell, Campinas, São Paulo, Brazil) supplemented with 10% fetal bovine serum (SFB) (Invitrogen, Waltham, MA, USA), 100 U $mL^{-1}$ penicillin, 100 U $mL^{-1}$ streptomycin and 10 mM 4-(2-HidroxiEthil)-1-PiperazinEthanolSulfonic buffer (HEPES) (*Mosmann, 1983*). Cells were cultured in sterile 96-well plates with level depths (Sarstedt, Nümbrecht, Germany) at a density of $5 \times 10^3$ cells per well and were then incubated in an incubator at $37 \pm 2$ °C in an atmosphere containing 5% $CO_2$ for 48 h. Subsequently, culture media were replaced with sample solutions at concentrations of 7.81–1,000 $\mu g$ $mL^{-1}$. Plates were then incubated at $37 \pm 2$ °C in an atmosphere containing 5% $CO_2$ for 48 h. Controls were not inoculated and contained 2 $\mu L$ of 70% (v/v) ethanol solution. After 48 h, culture media were removed

and 100 μL aliquots of 10% [3-(4,5-dimethylthiazol-2-yl)-2,5-diphenyltetrazolium bromide] (MTT; 5 mg mL$^{-1}$) in DMEM were added to all wells. Plates were immediately incubated at $37 \pm 2\,^{\circ}$C in an atmosphere containing 5% $CO_2$ for 3 h. Finally, the resulting formazan crystals were dissolved in DMSO and absorbance was evaluated using a spectrophotometer (SpectraMax 190; Molecular Devices, San Jose, CA, USA) at 540 nm (*Twentyman & Luscombe, 1987*).

### Statistical analysis

Data are expressed as mean ± standard errors of the mean and are representative of five replicates. Differences were identified using analyses of variance (ANOVA) followed by Bonferroni's test (Graphpad Prism version six and IBM SPSS Statistics 21 (SPSS, Inc., Chicago, IL, USA)) and were considered significant when $p < 0.05$.

## RESULTS AND DISCUSSION

The results show that this process allows for the rapid synthesis of AgNPs from the *C. ferrea* seed extract, which acts as a reducing and stabilizing agent for silver ions. AgNPs can be synthesized in minutes and up to 24 h in the presence of ammonia, which favors the formation of a soluble silver complex (*Chandran et al., 2006*; *Stephen & Seethalakshmi, 2013*).

In this study, AgNPs were synthesized in a few minutes and immediately followed by UV–Vis absorption spectroscopy and SEM. Figure 1 shows a UV–Vis absorption spectrum, after 24 h, in the region between 200 and 600 nm for a AgNO$_3$:NaOH solution (990:0.5 v/v) (black line), *C. ferrea* extract (red line) and AgNPs (blue line). The maximum absorption in the spectrum (blue line), centered at 423 nm, indicates the formation of AgNPs (the wavelength at 423 nm is a typical value) (*Seyed & Sattar, 2009*).

Figure 2 exhibits the SERS spectrum of R6G in aqueous 10 mmol L$^{-1}$ solution and dispersed in a colloidal suspension of AgNPs at $1.0 \times 10^{-5}$ mol L$^{-1}$ of the dye. R6G presents a very strong absorption band at 527 nm; therefore, to avoid the resonance Raman and fluorescence associated with R6G, exciting radiation at 633 nm was used. Both Raman and SERS spectra show the same spectral profile, containing similar vibrational features characteristic of the dye (*Hildebrand & Stockburger, 1984*) but with different relative intensities.

The adsorption of R6G on the silver surface enhances the intensity of some bands with a low cross section in the spectrum of the pure molecule. The change in relative intensity in SERS (*Zhou et al., 2012*) is a consequence of both chemical interaction of the dye with the surface and localized surface Plasmon resonance conditions, given by the local electric field enhancement near the metal surfaces because of the oscillating surface dipole (*Fan, Andrade & Brolo, 2011*). To verify the observed SERS enhancement on the AgNPs synthesized in the present work, we used the simplest equation for the SERS enhancement factor (*Le ru et al., 2007*), $EF_{SERS} = (I_{SERS}/I_{Raman}) \times (C_{Raman}/C_{SERS})$, where $I_{SERS}$ is the measured intensity for a given band, $I_{Raman}$ is the Raman intensity measured for the same band, $C_{Raman}$ is the concentration of R6G in the solution used for acquiring the Raman

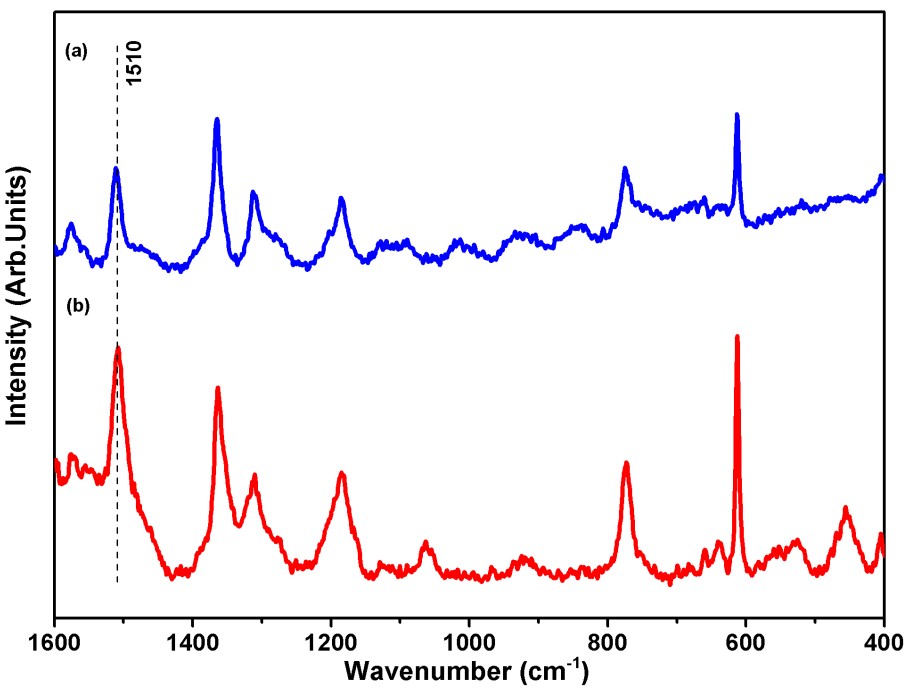

**Figure 2 Surface-enhanced Raman scattering (SERS) spectra of rhodamine-6G (R6G).** Raman spectra using laser excitation 633 nm of samples (A) 10 mmol $L^{-1}$ R6G and (B) silver nanoparticle $1.0 \times 10^{-5}$ mol $L^{-1}$ R6G.

spectrum, and $C_{SERS}$ is the concentration of R6G in the colloidal suspension. The band at 1,510 cm$^{-1}$ was chosen for the calculation. The synthesized AgNPs presented an EF$_{SERS}$ of approximately $7.2 \times 10^2$ times in the experimental conditions used, making AgNPs an interesting SERS substrate.

Figure 3 shows a SEM image and a histogram of the AgNP sizes obtained from the *C. ferrea* extract, always considering the longest axis for the measurement. SEM (Fig. 3A) confirmed the formation of AgNPs, and the predominant morphology is a spheroidal shape with a wide-size distribution. Furthermore, agglomerates are formed possibly by the drying process. In the histogram in Fig. 3B, the size distribution of the AgNPs shows a mean diameter approximately 30–50 nm. The histogram was obtained by counting approximately 140 particles in different regions. The morphology observed by SEM is consistent with the UV–Vis spectroscopy results presented in Fig. 1.

Figure 4 shows the XRD analysis of AgNPs. The diffractogram illustrates diffraction peaks at $2\theta = 38.15°, 44.25°, 64.47°, 77.38°$ and $81.64°$. The well-defined peaks indicate high crystallinity of the AgNPs, corresponding to a cubic unitary cell for nanocrystals. The UV–Vis spectroscopy, SERS, SEM and XRD results confirm the formation of AgNPs when using the *C. ferrea* extract as a reducing and stabilizing agent.

In relation to the antifungal activity the AgNPs were tested in the range of 9.7–5,000 μg mL$^{-1}$ (Table 1). It was observed antifungal action against *C. albicans* (MIC = 312.5 μg mL$^{-1}$), *C. glabrata* (MIC = 1,250 μg mL$^{-1}$), *C. kruzei* (MIC = 312.5 μg mL$^{-1}$) and *C. guilliermondii* (MIC = 156.25 μg mL$^{-1}$). Both *C. ferrea* extract and AgNPs were active under experimental conditions.

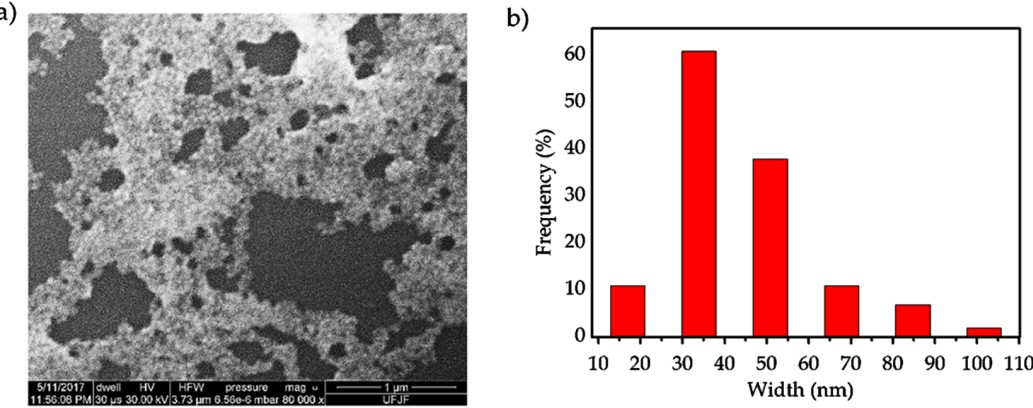

**Figure 3 Scanning electron microscopy (SEM).** (A) A representative SEM micrograph of the silver nanoparticles and (B) histogram of the size distribution of silver nanoparticles.

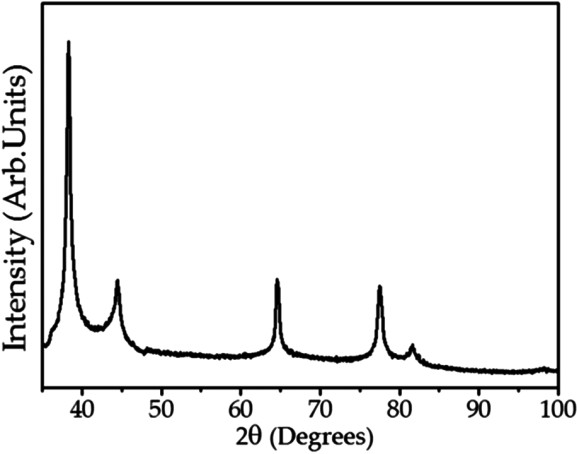

**Figure 4 X-ray diffraction (XRD) of the AgNPs.** X-ray diffraction related to the diffraction planes of silver nanoparticles.

The biocidal activity of AgNPs depends on their size, shape, surface coatings (*Wei et al., 2010*; *Zhang et al., 2016*) and ultrafine structure (*Harekrishna et al., 2009*). In a study by *Esteban-Tejeda et al. (2009)*, the synthesized AgNPs (20 nm) showed potent antifungal activity against clinical isolates and *Candida* ATCC standard strains in concentrations of 1–7 μg mL$^{-1}$.

*Prasad & Elumalai (2011)* and *Otunola et al. (2017)* used plant species for the synthesis of AgNPs and showed strong antimicrobial activity on pathogenic microorganisms, including *Candida tropicalis, C. krusei* and Gram-positive and Gram-negative bacteria. The synthesized AgNPs were spherical and between 3–22 nm and 57 nm in size.

Plant species have shown great potential in the accumulation of heavy metals and as bioreducing agents (*Iravani et al., 2014*) in addition to maintaining the antimicrobial activity of silver (*Alsalhi et al., 2016*). This action depends on the pH, reaction temperature and the nature and concentration of the extract (*Mittal et al., 2014*).

**Table 1 Minimum inhibitory concentration (MIC) and minimum fungicidal concentration (MFC) of *C. ferrea* extract, AgNPs and reference drugs.**

| Microorganisms | Nystatin | | Amphotericin B | | *C. ferrea* (Tul.) Martius | | AgNPs | |
|---|---|---|---|---|---|---|---|---|
| | MIC | MFC | MIC | MFC | MIC | MFC | MIC | MFC |
| *Candida albicans* ATCC 10231 | 0.331 | 3.3 | 0.125 | 0.5 | 9.7 | 1,250 | 312.5 | 2,500 |
| *Candida glabrata* CCT 0728 | 0.663 | 2.64 | 0.25 | 0.5 | 19.53 | 1,250 | 1,250 | 5,000 |
| *Candida krusei* CCT 1517 | 2.64 | 2.64 | 2 | 2 | 78 | >5,000 | 312.5 | 312.5 |
| *Candida guilliermondii* CCT 1890 | 1,326 | 10.56 | 0.0312 | 0.0312 | 39.06 | 156.25 | 156.25 | 312.5 |

**Note:**
Results of MIC and MFC were expressed in $\mu g\ mL^{-1}$.

Although the intracellular pathways involved for these biological effects still remain largely unclear, some authors highlight the presence of secondary metabolites, usually flavonoids, tannins and polyphenol compounds, as the cause (*Frasson, Bittencourt & Heinzmann, 2003*; *Carlson et al., 2008*; *Mittal et al., 2014*; *Alsalhi et al., 2016*; *Fatimah, 2016*).

Besides anthraquinones, alkaloids, depsides, depsidones, lactones, saponins, sugars, sesquiterpenes and triterpenes found in species of *C. ferrea* (*Souza et al., 2006*; *Sampaio et al., 2009*). These secondary metabolites may explain the positive action of the AgNPs reduced by the *C. ferrea* extract on most strains of *Candida* studied here: AgNPs effectively inhibited the growth of tested yeasts at concentrations below their cytotoxic threshold against the murine fibroblasts cells tested. The *C. ferrea* is an economical and ecological important specie in Brazil. NMR study revealed a galactomannan with a 2.65 mannose/galactose ratio was main seed polysaccharide of this plant species (*Gallão et al., 2013*). Further, the galactomannans, extracted from the endosperm of leguminous seeds, and responsible to perform functions of energy reservation and hydration. They have special properties such as high molar mass, water solubility and non-ionic character. A polysaccharide chain composed of at least 85–95% mannose units will provide intermolecular interactions like hydrogen bonds between *cis* hydroxyls of mannose, leading to formation of insoluble aggregates. In what concerns non-food industries, galactomannans are used as thickeners and stabilizers in pharmaceutical formulations, such as creams and lotions for cosmetic fields. In addition, these polysaccharides have been applied as matrix in the controlled release of drugs (*Albuquerque et al., 2016*).

From an ethanol extract of *C. ferrea* four phenolic compounds were isolated and identified by spectraldata: organic acid 1 and ester 2 (from green beans), biflavonoid 3 and phytoalexin 4 (from stem). Additionally, from this same extract, other constituents (two triterpenes, two steroids, two acid and fatty alcohol) were detected by gas chromatography and characterized by NMR. This plant species is well investigated under the point of view of biological activities, evidencing its therapeutic potential (*Magalhães et al., 2014*).

The study of *Sampaio et al. (2009)* showed that the extract of *C. ferrea* Martius has a good inhibitory effect upon oral microorganism since MIC values ranged between 250 and 1,000 $\mu g\ mL^{-1}$ which corroborates with the results of this study.

Extracts and polysaccharide fractions of *C. ferrea* pods exhibit potent anti-inflammatory activity via negative modulation of histamine, serotonin, bradykinin, PGE$_2$ and NO

could be interfering not only in the vascular, but also in cellular inflammatory events being well tolerated by animals (*Pereira et al., 2012*).

*Caesalpinia ferrea* has several biological activities (*Bariani et al., 2012*). With respect to cytotoxicity in murine L929 fibroblast cells exposed for 48 h to *C. ferrea* extract and coated AgNP, the results demonstrated a dose-dependent increase in AgNP concentration, causing significant cell death for the *C. ferrea* extract only at the highest concentration of 1,000 μg mL$^{-1}$.

*Hussain et al. (2005)* and *Alsalhi et al. (2016)* evaluated the in vitro toxicity of AgNPs of different sizes (15 or 100 nm) at different concentrations. The mitochondrial function decreased significantly in cells treated with higher doses of AgNPs, and the cells showed cell retraction and an irregular shape. In this study, at lower concentrations of AgNPs, greater cell viability is observed (3.5, 7.0 and 15 μg mL$^{-1}$ = 96.43% ± 0.43). Both the *C. ferrea* extract and AgNPs presented high cell viability in a percentage range of 82.32–102.71% and 67.13–96.43%, respectively, in concentrations ranging from 3.5 to 1,000 μg mL$^{-1}$ (Fig. 5).

Factors crucial for determining cytotoxicity include size, distribution, morphology and composition of AgNPs; coatings, agglomeration and rate of dissolution; reactivity of the particles in solution; and the type of reduction agents used for the synthesis of AgNPs. Of these, particle size is critical because of its influence on the bioavailability, biodegradability and in vivo toxicity of AgNPs, a factor that still needs to be better understood (*Wei et al., 2010*; *Marcato et al., 2012*; *Shinde et al., 2012*; *Zhang et al., 2016*).

*Jeeva et al. (2014)* showed that the morphology of the AgNPs synthesized using *Caesalpinia* leaf extracts was found as a triangle shape with a diameter range of 40–52 nm, whereas in the extracts of centrifuged leaves three triangle, hexagonal and spherical forms were observed with one average size between 78 and 98 nm. In addition, the synthesized AgNPs showed potential antimicrobial activity against human pathogens.

Small AgNPs have a large surface area, a greater potential to invade cells and, therefore, a greater cytotoxic potential (*Wei et al., 2010*; *Nogueira, Paino & Zucolotto, 2013*; *Zhang et al., 2016*). In a study by *Wei et al. (2010)*, AgNPs (50–100 nm) induced apoptosis, increased DNA fragmentation and induced structural mitochondrial and membrane damage in L929 cells, which resulted in cell death. These effects are because of the presence of silver ions, which deactivate cellular enzymes and damage the cell membrane of fungi. The main mechanism of action appears to be associated with increased levels of oxygen-reactive species (*Hussain et al., 2005*; *Shinde et al., 2012*; *Nogueira, Paino & Zucolotto, 2013*; *Monalisha, Swati & Nabin, 2014*; *Alsalhi et al., 2016*).

In this study, the presence of silver ions in AgNP coated with *C. ferrea* (30–50 nm) appeared to have no effect on L929 cells.

Green synthesis with *C. ferrea* provides advancement over chemical and physical method as it is cost effective, environment friendly, easily scaled up for large scale synthesis.

This study demonstrated that *C. ferrea* can be used as a bioreducing agent for the biosynthesis of AgNPs of adequate size, shape and stability. These AgNPs show

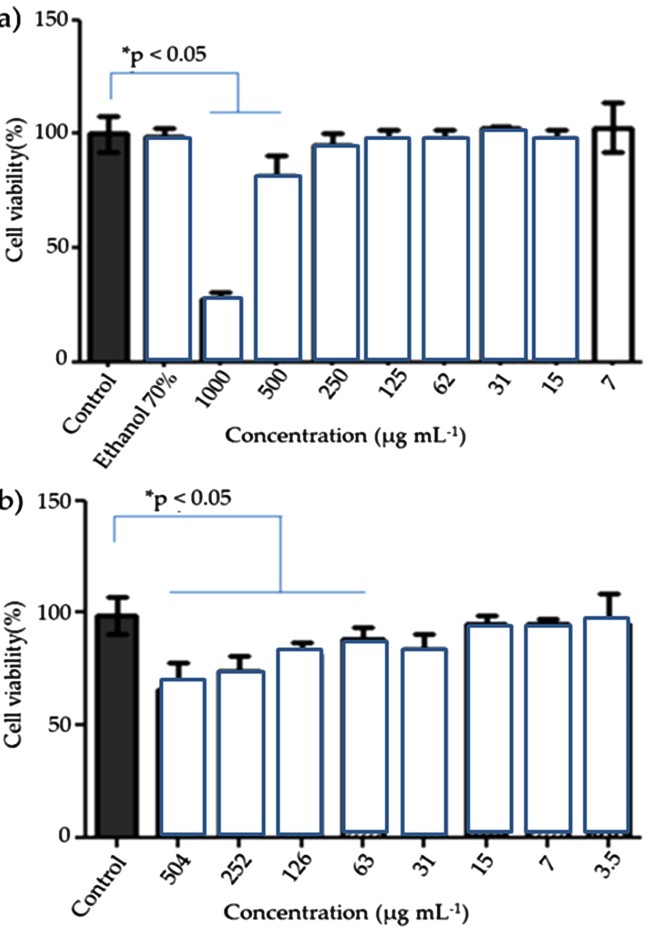

**Figure 5 Viability L929 fibroblast cells of the AgNPs.** Viability L929 fibroblast cells, 48 h post-exposure to extract of *C. ferrea* (Tul.) Martius (A) and AgNPs (B) in different concentrations. Data are expressed as mean ± standard errors of the mean *versus* control. Analyses of variance (ANOVA) followed by Bonferroni's test $^*p < 0.05$.                                             

considerable antifungal activity on *C. albicans, C. kruzei* and *C. guilliermondii* and high cell viability in murine L929 fibroblasts cells.

Our results should provide a valuable reference for the future treatment of fungal diseases. The ability to biosynthesize AgNPs using *C. ferrea* is highly promising as a simple and reproducible process of sustainable green synthesis. Nevertheless, additional in vivo studies should be undertaken to better understand the efficacy of AgNPs before developing them for clinical applications.

## CONCLUSION

This study demonstrated that *C. ferrea* can be used for AgNP biosynthesis. The biosynthesized AgNPs were adequate for size, shape and stability, obtained considerable antifungal activity on *C. albicans, C. kruzei* and *C. guilliermondii*. In addition, height cell viability on murine L929 fibroblasts cells. Therefore, the ability to biosynthesize AgNPs using *C. ferrea* is highly promising as a simple and reproducible process of sustainable green synthesis.

### Funding
The authors received no funding for this work.

### Competing Interests
The authors declare that they have no competing interests.

### Author Contributions

- Mônica R. P. S. Soares conceived and designed the experiments, performed the experiments, analyzed the data, contributed reagents/materials/analysis tools, prepared figures and/or tables, authored or reviewed drafts of the paper.
- Rafael O. Corrêa performed the experiments, contributed reagents/materials/analysis tools, authored or reviewed drafts of the paper.
- Pedro Henrique F. Stroppa performed the experiments, analyzed the data, contributed reagents/materials/analysis tools, prepared figures and/or tables, authored or reviewed drafts of the paper.
- Flávia C. Marques performed the experiments, contributed reagents/materials/analysis tools, authored or reviewed drafts of the paper.
- Gustavo F. S. Andrade performed the experiments, contributed reagents/materials/analysis tools, authored or reviewed drafts of the paper.
- Charlane C. Corrêa performed the experiments, contributed reagents/materials/analysis tools, authored or reviewed drafts of the paper.
- Marcos Antônio F. Brandão conceived and designed the experiments, performed the experiments, contributed reagents/materials/analysis tools, authored or reviewed drafts of the paper.
- Nádia R. B. Raposo conceived and designed the experiments, performed the experiments, analyzed the data, contributed reagents/materials/analysis tools, authored or reviewed drafts of the paper.

### Data Availability
Raw data can be found in the Supplemental Information.

### Supplemental Information
Supplemental information for this article can be found online at http://dx.doi.org/10.7717/peerj.4361#supplemental-information.

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
