# Peer review of "Biosynthesis of silver nanoparticles using Caesalpinia ferrea (Tul.) Martius extract: physicochemical characterization, antifungal activity and cytotoxicity"

_PeerJ, doi:10.7717/peerj.4361_

## Round 0.1 · original submission · Minor Revisions

Dear Dr. Raposo,

Although all 3 reviews are brief, please see below their comments which I will appreciate you carefully addressing. Thank you for a well-written manuscript and I look forward to the revised manuscript.

Best

Reviewer 1 ·

Basic reporting

-The article is written clear and unambiguous.
-Literature references are sufficient, however the discussion part need more relevant work to be addressed and compared.
- The structure of the article is acceptable with respect to the journal standard format.
-All results are presented well and they are relevant to the article hypothesis.

Experimental design

All experimental work is well designed.

Validity of the findings

All findings/results are novel.
Statistical results are presented well.

Additional comments

The author has to improve the discussion part, adding more relevant/new work with respect to the anti-fungal and cytotoxicity findings.

Reviewer 2 ·

Basic reporting

This manuscript is well written, at least, with professional English

Experimental design

The concept of silver nanoparticles is not new; however, the authors made use of C. ferrea seed extract for preparing such nanoparticles and they showed its biological application.

Validity of the findings

Nanotechnology is gaining importance and the present finding is timely in the context of nanotechnology.

Additional comments

The manuscript by Serna et al. is interesting and well written. However, much information are lacking: (i) the authors state that the C. ferrea seed extract is a reducing compound, evidence confirming its reducing property is lacking or elsewhere has it been shown? (ii) what is the composition of this C. ferrea seed extract? (iii) lines 311-318, authors do state that the secondary metabolites could be responsible for the fungicidal activity; however, what is the (at least speculated) mechanism of action during antifungal activity in the present experimental context?

Other comments:
1. Abstract: ……demonstrated that C. ferrea may be used for…., correct is at ……demonstrated that C. ferrea seed extract may be used for….
2. Line 216, …….in an oven at 37 ± 2ºC in an……., correct it as in an incubator at 37 ± 2ºC in an……..
3. In Figure 1 the authors do show the UV spectra for AgNO3:NaOH solution, but was it after incubating AgNO3:NaOH solution for 12-24 h?

Reviewer 3 ·

Basic reporting

The work is interesting and suitable for publication after minor revision and modifications.

Experimental design

Please explain in detail how the UV spectra were obtained. Was a colloidal suspension used? How was contribution from turbidity accounted for?

Validity of the findings

Baseline correction of spectra in figures 1 (UV-Vis) and Figure 2 (SERS) can help to better show the peak /location absorbance of the final product.

Why is there a loss in viability at higher concentrations? does the compound/product affect cell function? (Figure 5) . The plotting of the control and AgNP in the same figure could help to better show the effect of AgNP. What is the effect of uncoated AgNP?

Additional comments

Please check for the term oxygen-reactive species (Line344). Us it reactive oxygen species?

---

## Round 0.2 · Minor Revisions

Dear Dr. Raposo,

Thank you for your re-submission.

I went through the article and first of all, i noticed that you have submitted an article with the header and footer with "Int. J. Mol. Sci. 2017" and its "website". if this is a dual submission to both PeerJ and Int. J. Mol. Sci. please let us know. if it is not, please remove those. Please do not place any header and footer in your re-submission.

There are several articles showing similar applications as yours. Please highlight the novelty of your study over the others using Caesalpinia species (for eg., as in Journal of Applied Pharmaceutical Science Vol. 7 (08), pp. 226-233, August, 2017).

have you tested any other part of the plant C. ferrea. why only seeds? Does leaf extract, root extract have equal or better reducing capabilities? you can add a comment to this in your discussion if there are any references that support this.

Line 56: "green synthesis of AgNPs", please define green synthesis in methods, in both abstract and the main manuscript.

line 110: either 'have been' or 'was'

Thank you
best

---

## Round 0.3 · accepted · Accept

Dear Dr. Raposo,

Thank you for addressing the reviewers and editor's concerns and resubmitting the manuscript. I still see some spelling mistakes with regard to the organism names. Please be vigilant of that.

I am happy to recommend your article for publication in PeerJ. Congratulations.

Best
Priya Banada